# The Cynomolgus Macaque Intestinal Mycobiome Is Dominated by the *Kazachstania* Genus and *K. pintolopesii* Species

**DOI:** 10.3390/jof8101054

**Published:** 2022-10-08

**Authors:** Steve A. James, Aimee Parker, Catherine Purse, Andrea Telatin, David Baker, Sandy Holmes, James Durham, Simon G. P. Funnell, Simon R. Carding

**Affiliations:** 1Gut Microbes and Health, Quadram Institute Bioscience, Norwich Research Park, Norwich NR4 7UQ, UK; 2UK Health Security Agency, Porton Down, Salisbury SP4 0JG, UK; 3Norwich Medical School, University of East Anglia, Norwich NR4 7TJ, UK

**Keywords:** mycobiome, gastrointestinal tract, cynomolgus macaque, non-human primate, fungi, pathobiont, yeast, *Kazachstania pintolopesii*

## Abstract

The cynomolgus macaque, *Macaca fascicularis*, is a non-human primate (NHP) widely used in biomedical research as its genetics, immunology and physiology are similar to those of humans. They may also be a useful model of the intestinal microbiome as their prokaryome resembles that of humans. However, beyond the prokaryome relatively little is known about other constituents of the macaque intestinal microbiome including the mycobiome. Here, we conducted a region-by-region taxonomic survey of the cynomolgus intestinal mycobiota, from duodenum to distal colon, of sixteen captive animals of differing age (from young to old). Using a high-throughput ITS1 amplicon sequencing-based approach, the cynomolgus gut mycobiome was dominated by fungi from the Ascomycota phylum. The budding yeast genus *Kazachstania* was most abundant, with the thermotolerant species *K. pintolopesii* highly prevalent, and the predominant species in both the small and large intestines. This is in marked contrast to humans, in which the intestinal mycobiota is characterised by other fungal genera including *Candida* and *Saccharomyces,* and *Candida albicans*. This study provides a comprehensive insight into the fungal communities present within the captive cynomolgus gut, and for the first time identifies *K. pintolopesii* as a candidate primate gut commensal.

## 1. Introduction

The cynomolgus macaque (*Macaca fascicularis*), also known as the long-tailed or crab-eating macaque, is a cercopithecine primate indigenous to mainland Southeast Asia, as well as the maritime islands of Borneo, Java, and Sumatra, and islands of the Philippines. Like its close relative the rhesus macaque (*Macaca mulatta*), the cynomolgus macaque shares behavioural, immunological, and physiological similarities, as well as a close evolutionary relationship with humans, making this non-human primate (NHP) an important animal model for biomedical research [1]. These similarities appear to extend to the intestinal microbiota, with some bacterial taxa being common to both humans and non-human primates. The phyla Bacteroidetes, Firmicutes and Proteobacteria are prominent members of the prokaryome of both the human [2,3,4], and macaque gastrointestinal tract (GIT) [5,6,7,8]. Furthermore, a recent comparative metagenomic survey has shown that the gastrointestinal (GI) microbiota of cynomolgus macaques is more like that of humans than that of either mice or pigs [6]. Hence, these non-human primates represent a more suitable animal model for studying biological processes such as human ageing, and how age shapes the composition (and function) of the GI microbiota, and how this in turn affects GI physiology and host health [9,10].

Beyond the GI prokaryome however, relatively little is known of other constituents of the cynomolgus macaque GI microbiome and in particular, the fungal microbiome (mycobiome). Whilst typically present in low abundance [11], enteric fungi nevertheless interact with both the prokaryome as well as host cells to alter host immunity, and can exacerbate the severity of several human diseases, including inflammatory bowel disease (IBD) and colorectal cancer [12,13,14,15,16,17,18,19]. A recent study characterising the oral and faecal mycobiomes of wild and captive Thai cynomolgus macaques, represents the first and only such study to date [20], revealing wild macaques have a significantly higher fungal alpha diversity than their captive counterparts. Overall, most fungi in the faecal (and oral) mycobiome, which is a surrogate of the intestinal mycobiome, of these primates belonged to the Ascomycota phylum, with the cynomolgus faecal mycobiota dominated by the budding yeast genus *Kazachstania*. Thermotolerant members of this genus and those belonging to the *K. telluris* species complex (incl. *K. bovina*, *K. pintolopesii*, *K. slooffiae* and *K. telluris*), are often found in the GIT of cows, pigs and rodents [21,22,23,24,25,26].

In the present study, we employed a high-throughput internal transcribed spacer region 1 (ITS1) amplicon sequencing approach, using an established ITS1 primer set [27,28], to conduct a comprehensive region-by-region taxonomic survey of the cynomolgus macaque intestinal mycobiota, from duodenum to distal colon, in a cohort of young, adult, and aged captive animals. Our goal was to gain a better insight into the composition and diversity of the fungal communities populating the GIT of this biomedically important NHP species, investigate how they change with age and identify candidate fungal GIT commensals.

## 2. Materials and Methods

### 2.1. Animals

Sixteen clinically healthy cynomolgus macaques were included in the study. The animals ranged from 4 to 20 years in age and were categorized into young (<7 years), adult (8 to 12 years), or aged (13 years or older) (see Appendix A). All animals housed and bred at the UKHSA facility are derived from either Mauritian or South East Asia origin. No new animals have been introduced to these colonies since 2004. The colonies are licensed by the UK Home Office to breed, supply and use macaques for scientific research (Establishment license no. XBF9440B0). The breeding colonies are maintained to the highest standard in terms of animal welfare, health status, genetic profile, and behavioural compatibility, compliant with the UK Home Office Code of Practice for the Housing, and Care of Animals Bred, Supplied or Used for Scientific Purposes, 2014. This is achieved through facilities that provide an enriched and complex environment which meets the behavioural needs of the animals. The cynomolgus macaques are held in either harem breeding groups, or single sex, age matched holding groups. Their accommodation is a climate controlled, multiple room, solid floor caging system. Most groups also have access to an external ‘extension’ pen that is not climate controlled and open to the elements. All larger rooms have complex enrichment. Deep litter bedding is provided in the largest of these rooms. Water and a complete primate diet is provided ad lib. Fresh fruits, vegetables and pulses are provided daily as enrichment.

### 2.2. Sample Collection and DNA Extraction

All animals used in this study were required to be euthanized as part of normal colony management needs and requirements. Identified animals were initially sedated with ketamine hydrochloride at a dose of 10 mg/kg before exsanguination and euthanasia via intracardial injection with sodium pentobarbital at a dose of 80 mg/kg for elderly and 120–160 mg/kg for younger NHPs. All procedures were conducted under the authority and in compliance with a UK Homes Office project license. Luminal contents were collected from each GIT region of each animal and immediately frozen prior to transfer on dry ice to the laboratory for storage at −70 °C prior to processing. Total microbial DNA was extracted from ~200 mg of lumen content using the QIAamp PowerFecal Pro DNA kit (QIAGEN) and following the manufacturer’s protocol. In addition, all samples were homogenized using a FastPrep-24 benchtop tissue homogenizer (MP Bio) at 6.0 m/s for 1 min. This step was included to aid fungal cell wall disruption to improve fungal DNA recovery. Extracted DNA was quantified, and quality checked using the Qubit 3.0 fluorometer and associated Qubit dsDNA BR Assay Kit (Invitrogen). DNA samples were stored at −20 °C prior to further analysis.

### 2.3. ITS1 Amplification, Library Preparation and Sequencing

The fungal ITS1 region was amplified from 100 ng of template DNA by PCR using the ITS1F and ITS2 primer set [29,30], with each primer modified at the 5′ end to include an Illumina adapter tail using the following amplification conditions: 94 °C for 5 min; 35 cycles of 92 °C for 30 s, 55 °C for 30 s, and 72 °C for 45 s; and a final extension of 72 °C for 5 min. Amplification reactions were set up in duplicate for each faecal DNA sample, and positive and negative controls were also included in each PCR run (see Section 2.5). Following ITS1 PCR, a 0.7× SPRI purification using KAPA Pure Beads (Roche, Wilmington, MA, USA) was performed and the purified DNA was eluted in 20 µL of EB buffer (10 mM Tris-HCl). In a second PCR, library index primers were added using a Nextera XT Index Kit v2 (Illumina, Cambridge, UK) and amplified using the following conditions: 95 °C for 5 min: 10 cycles of 95 °C for 30 s, 55 °C for 30 s, and 72 °C for 30 s; and a final extension of 72 °C for 5 min. Following PCR, libraries were quantified using the Invitrogen™ Quant-iT dsDNA high sensitivity assay kit (Thermo Fisher, Waltham, MA, USA) and run on a FLUOstar Optima plate reader (BMG Labtech, Aylesbury, UK). Libraries were pooled following quantification in equal quantities. The final pool was SPRI cleaned using 0.7× KAPA Pure Beads, quantified on a Qubit 3.0 fluorometer and run on a High Sensitivity D1000 ScreenTape (Agilent Inc, Santa Clara, CA, USA) using the Agilent Tapestation 4200 to calculate the final library pool molarity. The pool was then run at a final concentration of 8 pM, on an Illumina MiSeq instrument using the MiSeq^®^ v3 (2× 300 bp) Kit (Illumina) at at the Quadram Institute Bioscience, Norwich. The raw data were analysed using MiSeq reporter. A mean sequence depth of 123,710 reads/sample was achieved; samples with fewer than 10,000 filtered sequences were excluded from further analysis (see Appendix A).

### 2.4. Mycobiome Characterization

Illumina MiSeq reads were analysed using the automated pipeline Dadaist2, a dedicated workflow for ITS profiling [31]. The quality profile of the raw reads (in FASTQ format) was assessed using SeqFu 1.9.3 [32], followed by primer removal using Cutadapt 3.5 [33] and quality filtering via Fastp 0.20.0 [34]. Locus-specific primers and conserved flanking regions were removed using ITSxpress [35]. The identification of representative sequences was performed using DADA2 [36], to produce a set of amplicon sequence variants (ASVs), and their taxonomic assignment was determined using the UNITE Fungal ITS database (release 8.3) [37]. The multiple alignment of the representative sequences was performed using ClustalO [38] and the guide tree was produced using FastTree [39]. Data normalization and diversity were produced using the Rhea scripts [40]. The output feature table, taxonomic classification, phylogeny and metafiles were exported and further analysed using PhyloSeq [41], MicrobiomeAnalyst [42], and the built-in plotting provided by Dadaist2 (via MultiQC [43]).

The raw Illumina ITS1 sequence data produced by the present study have been deposited at the European Nucleotide Archive (EBI), under the Project accession number PRJEB54860. Metadata and supporting scripts are available from the GitHub repository https://github.com/quadram-institute-bioscience/nhp-gut (22 July 2022).

### 2.5. Inclusion of Controls

Controls were included at each stage of the study. During DNA extraction, an empty bead-beating tube was included and treated the same as tubes containing luminal content and was quantified similarly. This extraction control was included in the initial amplicon PCR to assess that no ITS1 amplicon was produced. Negative (microbial DNA-free H_2_O) and positive controls (50 ng *K. telluris* DNA) were included in each PCR run. Libraries were also prepared from the DNA extraction control and from single fungal species DNAs (*C. albicans* and *K. telluris*) and were used as pipeline controls in the downstream bioinformatic analyses.

## 3. Results

### 3.1. Ascomycetous Fungi Dominate the Captive Cynomolgus Gastrointestinal Tract (GIT) Mycobiome

Fungal community profiling of the luminal contents of the duodenum to distal colon of a cohort of 16 captive macaques (NHP1 to NHP16) was performed using high-throughput internal transcribed spacer 1 (ITS1) amplicon sequencing and a DNA extraction protocol we developed and optimised to characterise the preterm infant GIT mycobiome [44]. The macaques ranged from 4 to 20 years in age and categorized into young (<7 years), adult (8 to 12 years), or aged (≥13 years) (Appendix A). A total of 6,927,777 quality trimmed ITS1 reads were obtained from 56 lumen samples, ranging from 14,599 (NHP 1, distal colon) to 248,538 (NHP7, caecum), with a sample average of 123,710 reads (Appendix A). Over 700 unique amplicon sequence variants (ASVs) were identified, although only 134 ASVs had a relative abundance of 0.01% or more. Collectively, this set of ASVs accounted for 99% of all ITS1 reads and was selected for subsequent taxonomic analyses to determine the composition and relative abundance of the fungal microbiota (mycobiota) in the separate intestinal sites of each macaque. The number of ASVs detected in each macaque ranged from 12 (NHP9) to 68 (NHP2), with over a quarter of ASVs specific to a single animal (36/134; 26.9%).

One hundred and thirty fungal taxa were classified to phylum level (97%), with the majority (95%) resolved to the genus level, and seventy-six of these (57%) to the species level. Four taxa could not be assigned at the phylum level or below and were classified as ‘Unidentified’ (Appendix A). At the phylum level, >99% of fungi belonged to either Ascomycota or Basidiomycota, with a single taxon assigned to the Mucormycota subphylum (Figure 1a)**,** with Ascomycota the predominant phylum accounting for 83% of all fungal reads (Figure 1a). In all, 85 taxa were ascomycetes, 44 were basidiomycetes, and one was identified as a mucormycete (*Mucor saturninus*) (Appendix A).

At the genus level, *Kazachstania* and *Debaryomyces* were the dominant genera, accounting for 76% of all ITS1 reads (Figure 1b). Both ascomycetous genera were detected, with varying abundance, in all macaque samples, irrespective of age. Overall, *Kazachstania* was the predominant genus, accounting for 49.0% of all ITS1 reads, and was present in varying abundance in all intestinal samples. Other notable, but less abundant genera included *Wallemia* (7.6%), *Scopulariopsis* (2.4%), and *Rhodotorula* (2.3%) (Figure 1b). Among the ten most abundant genera found in the macaque GIT, seven were yeast genera (*Candida*, *Cystobasidium*, *Debaryomyces*, *Filobasidium*, *Kazachstania*, *Rhodotorula* and *Symmetrospora*) (Appendix A).

### 3.2. Fungal Community Analysis and Identification of a Core NHP Gut Mycobiome

The 134 ASVs were also used to conduct a community analysis of the fungi present in each NHP age group. Specifically, to identify taxa (ASVs) that were specific to or found predominantly in a particular macaque age group, as well as those shared between two or more age groups. The analysis was restricted to those ASVs detected in at least 50% of animals in a particular age group (51/134; Appendix A). A Venn diagram was produced from the resulting dataset (Figure 2). In total, seventeen ASVs were found predominantly in the young macaques, seven in the adults, and one in the aged animals. Interestingly, a core set of thirteen ASVs, representing twelve different fungal taxa, were shared between all three age groups (Figure 2). Using type strain ITS1 sequences, ten taxa were resolved to species level and were identified as *Candida albicans*, *C. parapsilosis*, *Cutaneotrichosporon cutaneum*, *Debaryomyces hansenii*, *Filobasidium uniguttulatum*, *Kazachstania pintolopesii*, *Pichia fermentans*, *Rhodotorula mucilaginosa*, *Vishniacozyma carnescens* and *Wallemia muriae* (Appendix A). Amongst these fungi, only *C. albicans*, *C. parapsilosis*, *K. pintolopesii* and *P. fermentans* are known to grow at 37 °C (or above), and thus represent candidate cynomolgus gut commensals. Overall, two species, namely *D. hansenii* and *K. pintolopesii*, were found in every single macaque, irrespective of animal age (Appendix A).

### 3.3. Kazachstania pintolopesii and Debaryomyces hansenii Are Prevalent throughout the Cynomolgus GIT

Within the enteric fungal communities, two yeast species, *Kazachstania pintolopesii* and *Debaryomyces hansenii* (*Candida famata*) dominated and were present throughout the macaque GIT. Both species were present in all macaque samples irrespective of age, and with varying abundance (Figure 3). Overall, *K. pintolopesii*, a thermotolerant yeast species [22] dominated most GIT samples of adult and aged macaques (72.7%), and caecum of one young macaque (NHP9) (Figure 3). In contrast, *D. hansenii*, a yeast widespread in nature [45], dominated the GIT of two young macaques (NHP11 and NHP12) and most intestinal samples (9/11) of three aged animals (NHP1, NHP7 and NHP10) (Figure 3).

Although luminal samples could not be obtained from each intestinal region of every animal, samples (40) from nine animals (2 young, 3 adult and 4 aged) were used to conduct a comparative analysis of the most abundant fungi in the small and large intestine. Duodenum-, jejunum- and ileum-derived samples were pooled for taxonomic profiling of the small intestine, while the caecal and colon content samples were likewise pooled for the large intestine analysis. Overall, the two fungal profiles were broadly similar, with *K. pintolopesii* and *D. hansenii* being the most abundant species in both intestinal regions (Figure 4). Two other fungi present in both intestinal regions, but in lower abundance, were the basidiomycetes *Wallemia muriae* and *Rhodotorula mucilaginosa*. Notable differences between the two intestinal regions included the presence of *Scopulariopis brevicaulis*, a soil saprotroph, in the small intestine (Figure 4a), and presence of an *Aspergillus piperis*-like species in the large intestine (Figure 4b).

The caecum provided the most samples (13/16 macaques) (Appendix A), enabling a detailed comparative characterization of the fungal communities of this section of the macaque GIT. Three distinct profiles (C1–3) were identified based upon the presence and relative abundance of *D. hansenii* and *K. pintolopesii* (Figure 5). In the majority (C1; 7/13 macaques), *K. pintolopesii* was the predominant species. The second commonest (C2; 5/13) was *D. hansenii*, with a third profile (C3) characterized by the presence of basidiomycetous taxa (e.g., *Cystobasidium pallidum* and *Rh. mucilaginosa*). The C3 profile was atypical and restricted to a single adult female (NHP2) (Figure 5). In this animal, this species profile was caecum-specific, and was not replicated in either of the other two intestinal sites analysed, *D. hansenii* being predominant in the duodenum and *K. pintolopesii* in the ileum (Figure 3).

### 3.4. Prevalence of Human-Associated Fungi

*Candida* such as *C. albicans* and *C. parapsilosis* are frequent members of the human GIT mycobiota [44,46,47,48]. In our captive macaque cohort this genus accounted for <1% of all fungal reads (Appendix A). A total of eight *Candida* species were identified, including *C. albicans*, *C. parapsilosis* and *C. tropicalis*, all of which are fungal pathobionts [49]. In contrast, *C. anglica*, *C. freidrichii*, *C. oleophila*, *C. saitoana* and *C. sake* were considered GI transients based on their inability to grow above 30 °C [49].

*Candida parapsilosis* was the most prevalent *Candida* sp. and except for NHP8 was detected in all animals. *C. albicans*, while less prevalent (12/16) was present in all age groups (Figure 6). For those animals for which multiple intestinal samples were available (Appendix A) these two *Candida* were not restricted to one region of the GIT (e.g., *C. albicans* in NHP14; *C. parapsilosis* in NHP12) (Figure 6). In contrast, *C. tropicalis* was detected in only 4 GIT samples of three aged animals (NHP7, NHP15 and NHP16; Figure 6).

*Saccharomyces cerevisiae* frequently found in the human GIT [47], acquired via diet rather than by vertical transmission [50], accounted for less than 0.2% of all ITS1 reads in NHP samples. This species was detected in varying abundance (0.02 to 4.2%; Appendix A) in only seven macaques; two young (NHP9 and NHP12), two adult (NHP2 and NHP13) and three aged (NHP7, NHP10 and NHP15). Like *Candida* spp., *S. cerevisiae* was not restricted to one GI region and in NHP12 was detected in all three small intestinal sites, as well as the caecum.

*Debaryomyces hansenii*, a food-borne (dairy) yeast commonly detected in the human GIT [45,47] was the most prevalent and abundant human-associated fungus identified in this study. This yeast was detected in all macaques and was present in moderate to high abundance in five macaques; two young (NHP11 and NHP12) and three aged (NHP1, NHP7 and NHP 10) (Figure 3; Appendix A).

## 4. Discussion

Non-human primates, such as cynomolgus macaques represent important animal models in microbiome research, not least for the unprecedented opportunity they offer for gaining better insights into the biological processes (e.g., ageing) and factors (e.g., diet) that influence and shape the human microbiome. To date, much of this research has concentrated on the NHP prokaryome, with relatively little attention given to the other constituents, including intestinal fungi which are typically present in the GIT in much lower abundance than their bacterial counterparts [11]. To begin addressing this shortfall, we used a high-throughput ITS1 amplicon sequencing approach, that we first developed and used to profile the preterm infant GIT mycobiome [44], to characterize the enteric mycobiota in lumen contents collected from six intestinal sites of a cohort of captive macaques of differing age (i.e., from young to old). The results revealed that the cynomolgus GIT, from duodenum to distal colon, is populated by more than 50 genera, almost exclusively from the Ascomycota and Basidiomycota phyla, with Ascomycota the predominant phylum. This dominance, in both the macaque small- and large intestine, was largely due to the presence of taxa from the budding yeast genera *Debaryomyces* and *Kazachstania*, which together accounted for >70% of fungal reads in each intestinal region.

*Kazachstania* is the predominant genus in our captive macaque cohort, accounting for >72% of all fungal reads, and was found with varying abundance in every animal, and across all age groups. The dominance of this ascomycetous genus was due almost exclusively to one species, *K. pintolopesii*. While *K. telluris*, a close relative, was found in a limited number of macaques from each age group, it was typically present in very low abundance (<1.0%). In contrast, *K. pintolopesii* was highly prevalent throughout the small- and large intestines of each macaque. Furthermore, it was frequently the predominant fungus, and in a third of all GIT samples, *K. pintolopesii* abundancy exceeds 90%.

*Kazachstania* is a large and diverse yeast genus comprising more than 40 species [51]. Within the genus, *K. pintolopesii* is closely related to *K. bovina*, *K. heterogenica*, *K. slooffiae* and *K. telluris*. Collectively, these five comprise the *K. telluris* species complex, a phylogenetically distinct group of yeasts characterized by their ability to grow at elevated temperature (i.e., 37 °C) [22]. Some representatives of *K. pintolopesii* can survive and grow at temperatures as high as 42 °C [22], a physiological trait rare in yeasts. To date, most strains from this thermotolerant species complex have been isolated from the nasal passages and GIT of birds and mammals [21,22,23,24,25]. Prior to this study, the principal hosts of *K. pintolopesii* appeared to be mice (captive and wild) and rats [21,22,25] with the only member of the *K. telluris* species complex previously found in NHPs being *K. heterogenica*, which was limited to a single strain from a young female white-handed gibbon (*Hylobates lar*) [52].

*Debaryomyces hansenii* is also highly prevalent in the captive macaque cohort. Distributed throughout the cynomolgus GIT it was the predominant fungus in both the young and elderly animals, albeit to a lesser extent than *K. pintolopesii*. Despite having a lower optimum growth temperature than *K. pintolopesii* [22,45], this halotolerant food-borne (dairy) yeast is a frequent member of the human GI mycobiome [44,48,53,54] and can be cultured from human faeces, and is associated with ulcerative colitis (UC), Crohn’s Disease (CD) and colorectal cancer [19,55,56]. *D. hansenii* produces mycocins that kill *C. albicans* [57] and it was interesting to note that in three macaques where this yeast was in high abundance, *C. albicans* was absent. However, given the low abundance (<1.0%) of *C. albicans* in the macaque GIT it is difficult to draw any significance from these observations.

The human GIT like that of the cynomolgus macaque, is largely dominated by fungi from the Ascomycota and Basidiomycota phyla [20,46,48,53,58,59]. However, despite this similarity, our study has revealed distinct differences at both the genus and species levels between the intestinal mycobiota in humans and cynomolgus macaques. Most notable is the predominance of the *Kazachstania* genus in the cynomolgus gut, with *K. pintolopesii*, a fungus rarely found in humans, the dominant species throughout the cynomolgus GIT (this study; [20]). In contrast, *Candida* and *Saccharomyces* are prominent genera in the human intestinal mycobiome [46,48,53,58,59], often attributed to the presence of the fungal pathobiont *C. albicans* and the food-borne yeast *S. cerevisiae* [46,48,53,58,59]. Despite being rare in the environment [60], *C. albicans* is a common commensal of both the human GIT and oral cavity, and of the vaginal mycobiome [44,48,53,61,62,63]. Although human-associated fungi were detected in the cynomolgus GIT, most in low abundance (e.g., *C. albicans, C. parapsilosis* and *S. cerevisiae*) with the exception of *D. hansenii*, which was highly prevalent in the macaque cohort, and the predominant intestinal fungus in some of the young and aged animals. In addition to its use as a dairy yeast [45], *D. hansenii* is ubiquitous in nature, and frequently isolated from soil [45,64]. Thus, the presence of this ascomycete in the cynomolgus GIT may be of environmental origin but not from the primate diet. In contrast, the presence of *P. fermentans*, which was detected in many of the macaques, albeit in low abundance, is most likely due to diet. This yeast is frequently found in food and fruit juices [65], and so was most likely acquired from the daily supplement of fresh fruits and vegetables given to the animals. Two basidiomycetous yeasts that were also frequently detected were *Cutaneotrichosporon cutaneum* and *Filobasidium uniguttulatum*. *Cut. cutaneum*, like other members of this genus, is a common colonizer of animal skin [66], and its presence and prevalence could be as the result of skin-oral contact (i.e., grooming) within the colony. Indeed, in Thai cynomolgus macaque, Sawaswong and colleagues found *Cutaneotrichosporon* to be the prominent fungal genus of the oral microbiome of captive macaques [20], providing further support for social grooming as an additional route of fungal acquisition and transmission between primates. *F. uniguttulatum*, like *Cut. cutaneum*, is unable to grow at elevated temperature (i.e., 37 °C), and although its natural habitat remains unknown, it has been isolated previously from animal bedding [67]. Thus, given that the macaques are provided with deep litter bedding this may explain why this basidiomycete was detected in many of the animals. In addition to diet and environment, other factors that may contribute to help shape the enteric fungal communities in cynomolgus macaques and humans, include differing GI physiology and normal core body temperature (i.e., humans, 37.0 °C; macaques, 37.0 to 39.5 °C) [68].

The persistence of *K. pintolopesii* throughout the GIT in young, adult as well as elderly cynomolgus macaques coupled with an innate ability to grow at and above 37 °C [21], suggests that *K. pintolopesii* represents a plausible primate GI commensal. If proven then this raises the question as to what role it performs in the cynomolgus GIT. Insights into its potential role(s) may come from *K. slooffiae*, a close relative, and the predominant fungus in the post-weaning porcine gut [21,23,24,69]. This member of the *K. telluris* species complex [21,22] provides amino acids as an energy source for microbial and piglet growth and is an important source of health promoting micronutrients including vitamin C and formic acid [23,26,69]. Furthermore, a strong (positive) correlation has been identified between *K. slooffiae* and beneficial intestinal bacteria, including *Lactobacillus* and *Prevotella* [24]. In the human GIT, *C. albicans* can interact directly with *Lactobacillus* spp. [70,71], leading to the proposal that *K. slooffiae* may behave similarly to commensal *Candida* spp. in humans [24]. Given the paucity of human-associated *Candida* (e.g., *C. albicans* and *C. parapsilosis*) in these macaques, coupled with the prevalence and abundance of *Prevotella* and *Lactobacillus* spp., including *L. acidophilus* and *L. reuteri*, in the cynomolgus GIT [6], it is conceivable that *K. pintolopesii* performs an equivalent role to that of *K. slooffiae* in pigs. However, this remains to be established in future cynomolgus microbiome studies encompassing both the mycobiome and bacteriome.

Finally, *C. albicans* is a common member of the normal human microbiome, and in healthy individuals, it can remain a lifelong benign commensal. However, under certain circumstances (e.g., immunosuppression, broad-spectrum antibiotic treatment) it can cause infections ranging from superficial infections of mucosal surfaces to life-threatening systemic candidiasis [62,72,73]. *C. albicans* outgrowth in the human GIT can also compound pre-existing CD and UC [18,74]. Thus, given the pathobiont nature of *C. albicans* (in humans), future research is needed to investigate the pathogenic potential of *K. pintolopesii* in the cynomolgus macaque, and factors that may trigger a transitional shift from harmless commensal to pathogen. This is especially pertinent given that *K. pintolopesii* causes gastric infections in laboratory mice, which can prove fatal [21]. Moreover, *K. heterogenica*, another member of the *K. telluris* species complex and close relative of *K. pintolopesii* [22], can exacerbate *Helicobacter suis*-associated gastric infection in Mongolian gerbils [75], and has been linked to a fatal infection in a young female white-handed gibbon, the first documented case of its kind [52].

## 5. Conclusions

Our study identified a diverse array of fungi throughout the cynomolgus GIT, with Ascomycota and Basidiomycota the dominant phyla. A characteristic feature is the prominence of the ascomycetous yeast *K. pintolopesii*, a member of the *K. telluris* species complex, which we propose represents a credible primate intestinal commensal. This study paves the way for further investigations, firstly to confirm that *K. pintolopesii* is a primate GIT commensal, and if proven then establish what function it performs in the cynomolgus macaque GIT.

## Figures and Tables

**Figure 1 jof-08-01054-f001:**
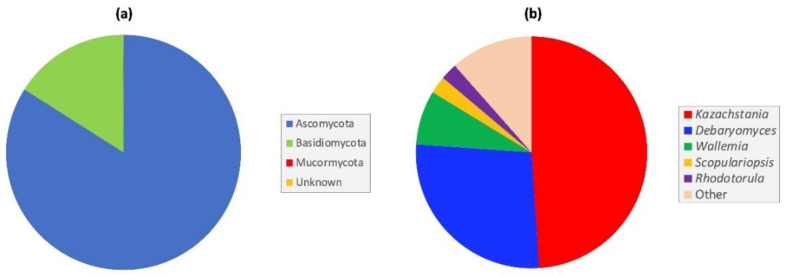
Most abundant fungi in the captive cynomolgus macaque GIT at (**a**) phylum, and (**b**) genus level.

**Figure 2 jof-08-01054-f002:**
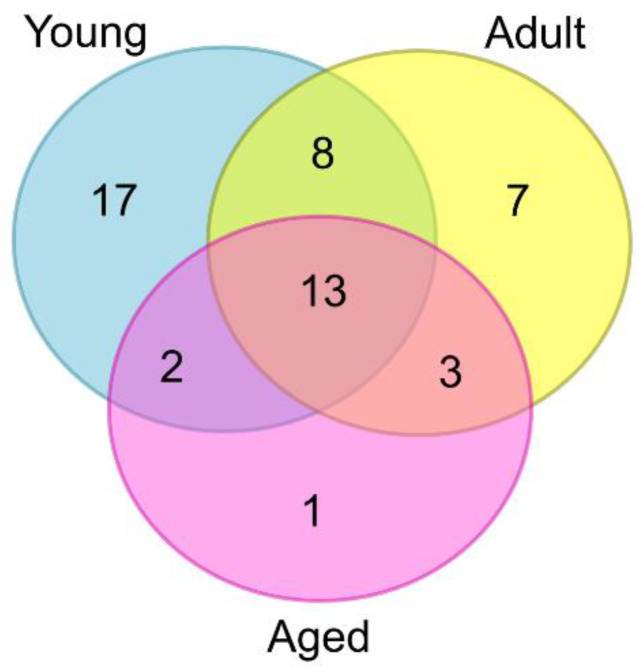
Venn diagram showing the number of fungal taxa (ASVs) found predominantly in each macaque age group (diagram produced using web-based software at: https://www.vanderpeerlab.org/?q=tools/venn-diagram (16 September 2022).

**Figure 3 jof-08-01054-f003:**
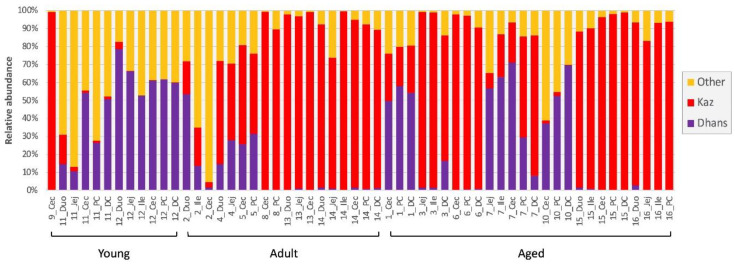
Prevalence and abundance of *K. pintolopesii* (Kaz) and *D. hansenii* (Dhans) in macaque GIT samples.

**Figure 4 jof-08-01054-f004:**
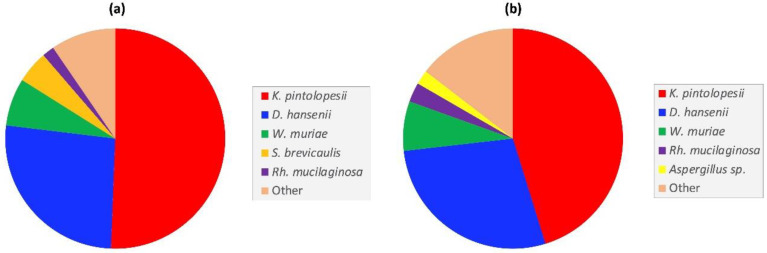
Comparison of the most abundant fungal taxa in the captive macaque (**a**) small- and, (**b**) large intestines.

**Figure 5 jof-08-01054-f005:**
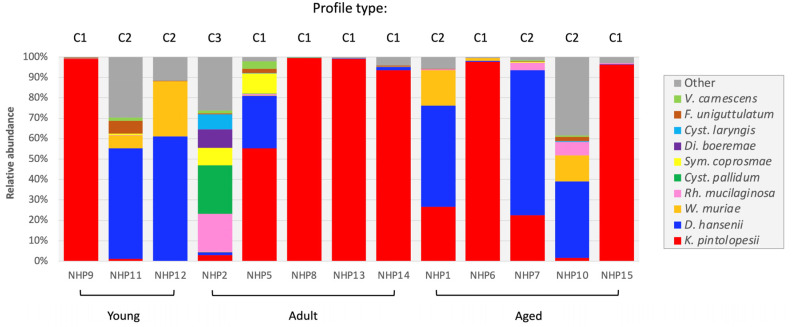
Comparison of the caecal mycobiome of 13 captive macaques of differing age (young, adult or aged).

**Figure 6 jof-08-01054-f006:**
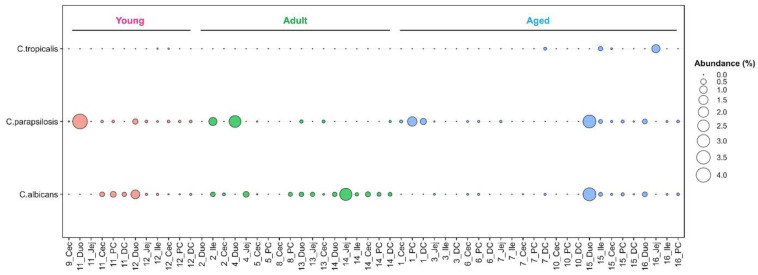
Prevalence and abundance of *Candida* pathobionts in the captive macaque GIT.

## Data Availability

The raw Illumina ITS1 sequence data used in this study have been deposited at the European Nucleotide Archive (EBI), under the Project accession number PRJEB54860. Metadata and supporting scripts are available from the GitHub repository https://github.com/quadram-institute-bioscience/nhp-gut, accessed on 22 July 2022.

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
