# Peer review of "The Cynomolgus Macaque Intestinal Mycobiome Is Dominated by the Kazachstania Genus and K. pintolopesii Species"

_jof, 2022, doi:10.3390/jof8101054_

Round 1

Reviewer 1 Report

The intestinal microbiome includes several types of organisms.    The microbiome is largely bacteria with Arachae, however, viruses are present in large quantities as well as mycotic organisms.  The present study uses correct and standard sequencing technologies to determine the major mycotic organisms in longitudinal sections of the non human primate macaques.  This is a very focused study and  presents the questions that  the reader will want to know more about. First, what are the differences in the non human primate that make their intestinal mycobiome different than humans.  Second, what differential functions might this non human primate mycobiome serve.  Third, as a human I am interested in the pathobiont mycotic Candida and why macaques do not contain this important organism. 

            These studies are interesting and of potential importance, however presently there is modest amounts of data.   The authors have been conservative to interpret the data and they do raise some of the important questions, however, there are instances where data can be obtained of importance and interest.  One item would be to analyze the stool mycobiome from different non human primates such as chimpanzees, benobos, oranguatans, gorillas and others.  It would be interesting to determine if there were phylogenetic determination of the intestinal mycobiome or infact the bacterial or viral intestinal microbiome.  The investigations might feed the macaques different diets and determine effects on the mycobiome.   Studies might be done on macaques from different regions to investigate local environmental factors that influce the mycobbiome.   Stool from different seasons might be tested and other conditions that might be important. Any of these would increase the impact of the studies.  

Author Response

Dear editors and reviewers,

We would like to thank the reviewers for their helpful, constructive comments and suggestions. We have made the necessary changes to the text which are indicated by Track Changes. We have also included an additional figure (Figure 2; Venn diagram) and provided additional data in the form of Supplementary Table S5 (re. fungal community analysis outputs). We hope that we have satisfactorily addressed all the reviewers’ queries and we include our point-by-point response below.

Reviewer #1

These studies are interesting and of potential importance, however presently there is modest amounts of data. The authors have been conservative to interpret the data and they do raise some of the important questions, however, there are instances where data can be obtained of importance and interest. One item would be to analyze the stool mycobiome from different non-human primates such as chimpanzees, benobos, oranguatans, gorillas and others. It would be interesting to determine if there were phylogenetic determination of the intestinal mycobiome or in fact the bacterial or viral intestinal microbiome. The investigations might feed the macaques different diets and determine effects on the mycobiome. Studies might be done on macaques from different regions to investigate local environmental factors that influence the mycobiome. Stool from different seasons might be tested and other conditions that might be important. Any of these would increase the impact of the studies.

The specific focus of our study was the captive cynomolgus macaque. Given its role as an important animal model in biomedical research, our aim was to conduct a comprehensive taxonomic survey of the intestinal mycobiota of this NHP, which to date, has been largely overlooked in NHP microbiome research. This study builds (and expands) upon the previous work by Sawasong and colleagues (cited). By targeting the ITS1 region we have also been able to resolve key fungal taxa to the species level, notably Debaryomyces hansenii and Kazachstania pintolopesii, with the latter identified, for the first time, as a candidate primate gut commensal. Whilst the suggested lines of investigation would indeed be interesting (and informative) they are clearly beyond the scope, scale and indeed budget of our current study.

Reviewer 2 Report

COMMENTS TO THE MANUSCRIPT “The Cynomolgus Macaque Intestinal Mycobiome is Dominated by the Kazachstania Genera and K. pintolopesii Species” by James et al.

General comment:

The fungal diversity (mycobiome) of the gastrointestinal tract of cynomolgus macaque (Macaca fascicularis) was analyzed by high-throughput ITS1 amplicon sequencing approach. Mycobiome was analyzed in three age groups (young, adult aged) through six gastrointestinal tract sections from small intestine (duodenum, Jejunum, Ileum) to large intestine (Caecum, Proximal colon, Distal colon). A total of 134 amplicon sequence variants (ASVs) that accounted for 99% of the ITS reads were analyzed. At phylum level, Ascomycota and Basidiomycota accounted for >99% of all reads, with 85 taxa identified as ascomycetes, 44 as basidiomycetes, one as a mucormycete, and four remained as Unidentified. Kazachstania and Debaryomyces ascomycete yeast genera were the predominant taxa. All manuscript sections are clearly explained, and the obtained results are properly described and discussed. The subject of the submitted manuscript is of interest for mycologists, and biomedical scientists studying microbiome in animal models. The manuscript is suitable to be published in the Journal of Fungi. Below are some specific comments for the authors' consideration.

Specific comments:

1. Besides describe the abundance of fungal genera/species you have enough data to conduct a descriptive fungal community analysis showing your results in a friendly graphical manner like Venn Diagram or Principal component analysis. This kind of analysis can give further insight into the fungal community composition differences at genera and species level between cynomolgus macaque age groups and/or gastrointestinal tract sections (see examples on the use of these tools for microbiome analysis in: Su et al. 2020. MicrobiologyOpen, 9(6): 1085-1101; Frontiers Microbiol. 2019. 10: 1263; Duan et al. 2021. Aquaculture Reports, 20: 100742). Furthermore, Venn diagrams can help you to find the core gastrointestinal mycobiome of cynomolgus macaque (Shade and Handelsman, 2012. Environ. Microbiol. 14(1): 4-12). You can generate Venn diagrams using friendly web servers (http://bioinformatics.psb.ugent.be/webtools/Venn/; http://www.interactivenn.net; https://www.biovenn.nl/). I think the impact of your manuscript will be higher if you include such analysis.

2. You first define the GIT acronym for the gastrointestinal tract (line 41), but through the text you use indistinctly GIT or GI. Please be consistent in the use of the acronym once defined.

Author Response

Dear editors and reviewers,

We would like to thank the reviewers for their helpful, constructive comments and suggestions. We have made the necessary changes to the text which are indicated by Track Changes. We have also included an additional figure (Figure 2; Venn diagram) and provided additional data in the form of Supplementary Table S5 (re. fungal community analysis outputs). We hope that we have satisfactorily addressed all the reviewers’ queries and we include our point-by-point response below:

Reviewer #2

  1. Besides describe the abundance of fungal genera/species you have enough data to conduct a descriptive fungal community analysis showing your results in a friendly graphical manner like Venn Diagram or Principal component analysis. This kind of analysis can give further insight into the fungal community composition differences at genera and species level between cynomolgus macaque age groups and/or gastrointestinal tract sections (see examples on the use of these tools for microbiome analysis in: Su et al. 2020. MicrobiologyOpen, 9(6): 1085-1101; Frontiers Microbiol. 2019. 10: 1263; Duan et al. 2021. Aquaculture Reports, 20: 100742). Furthermore, Venn diagrams can help you to find the core gastrointestinal mycobiome of cynomolgus macaque (Shade and Handelsman, 2012. Environ. Microbiol. 14(1): 4-12). You can generate Venn diagrams using friendly web servers (http://bioinformatics.psb.ugent.be/webtools/Venn/;http://www.interactivenn.net; https://www.biovenn.nl/). I think the impact of your manuscript will be higher if you include such analysis.

We have now conducted a community analysis to identify the fungal taxa (ASVs) that are found predominantly in each age group (i.e., young, adult and aged), as well as those that are shared between all three age groups (i.e., the ‘core’ gut mycobiome). The results of this analysis are discussed in the revised Results section (3.2. ‘Functional community analysis and identification of a core NHP gut mycobiome’) and presented as a Venn diagram (new Figure 2). Furthermore, from this analysis we have also been able to identify a subset of fungi (4 taxa) that are candidate gut commensals, based on their ability to grow at 37°C. This is now presented as additional data in Supplementary Table S5 (new).

  1. You first define the GIT acronym for the gastrointestinal tract (line 41), but through the text you use indistinctly GIT or GI. Please be consistent in the use of the acronym once defined.

We have now defined the GI acronym for gastrointestinal (line 43) and use each acronym (GI and GIT) accordingly throughout the text.

Reviewer 3 Report

In this study, the authors present a study of the mycobiome of the Cynomolgous Macaque. The literature is lacking on the mycobiome in non-human primates, and this study begins to address this shortfall. Overall, the manuscript is well-written and clearly presented. The authors include sufficient detail on the methods and included the information for the publicly-accessible raw data. Detailed comments below:

-The authors state that K. pintolopesii is a primate gut commensal, but this commensal state was not explicitly tested or proven. The authors should state that it is a "proposed" commensal until those studies can be performed.

-In Figure 3a, this data is described as the "small intestine". Just to clarify, is this the pooling of the duodenum, jejunum, and ileum? Or a subset of those?

-One caveat of this study that was not discussed was the fact that fungal studies have some limitations in that the choice of primer target region (18S, ITS1, or ITS2) can vastly change the fungal population found. The same is true with DNA extraction protocols. While I am not suggesting doing further sequencing, etc. The discussion would benefit from a short section discussing the author's knowledge of this and a justification for the choice of ITS1 for this study in the Cynomolgus Macaque. 

-Figure legends and axis labels may benefit from a larger font for easier audience reading.

Author Response

Dear editors and reviewers,

We would like to thank the reviewers for their helpful, constructive comments and suggestions. We have made the necessary changes to the text which are indicated by Track Changes. We have also included an additional figure (Figure 2; Venn diagram) and provided additional data in the form of Supplementary Table S5 (re. fungal community analysis outputs). We hope that we have satisfactorily addressed all the reviewers’ queries and we include our point-by-point response below.

Reviewer #3:

  1. The authors state that K. pintolopesii is a primate gut commensal, but this commensal state was not explicitly tested or proven. The authors should state that it is a "proposed" commensal until those studies can be performed.

Kazachstania pintolopesii is now identified as a ‘candidate’ primate gut commensal, and we acknowledge, in both the Results and Discussion sections, that further investigations will need to be conducted to resolve/confirm its status (as a primate gut commensal).

  1. In Figure 3a, this data is described as the "small intestine". Just to clarify, is this the pooling of the duodenum, jejunum, and ileum? Or a subset of those?

Yes, the data for the small intestine (Fig. 3a) was indeed pooled from luminal samples of the duodenum, jejunum and ileum, and we have now clarified this point in the text. Likewise, the data for the large intestine (Fig. 3b) was pooled from the caecal and colonic luminal samples, and this too is now clarified in the text.

  1. One caveat of this study that was not discussed was the fact that fungal studies have some limitations in that the choice of primer target region (18S, ITS1, or ITS2) can vastly change the fungal population found. The same is true with DNA extraction protocols. While I am not suggesting doing further sequencing, etc. The discussion would benefit from a short section discussing the author's knowledge of this and a justification for the choice of ITS1 for this study in the Cynomolgus Macaque.

In this study, we used a protocol that we have developed and optimised to characterise the preterm infant gut mycobiome. The results from this infant study were published recently in JoF (doi:10.3390/jof6040273), and we refer to this study and the protocols employed, in the current work (Ref # 42). In the Introduction, we now also highlight the fact that fungi are typically present in the GIT in low abundance and have included an additional reference to support this. Furthermore, in Section 2.2 (‘Sample collection and DNA extraction’) we now provide a short explanation as to why we included a bead-beating step in our DNA extraction protocol (i.e., to aid/enhance fungal cell wall disruption and thereby improve fungal DNA recovery).

Whilst we accept that there is recognised amplification bias associated with each ITS region (i.e., ITS1, basidiomycete-bias; ITS2, ascomycete-bias), a topic well-documented (e.g., Bellemain et al. 2010; BMC Microbiology 10_189), we believe we have provided sufficient explanation for the experimental rationale we have used in the current cynomolgus study.

  1. Figure legends and axis labels may benefit from a larger font for easier audience reading.

We have increased the font size for the figure legends and axis labels as requested, and trust they are now easier to read.

Round 2

Reviewer 1 Report

The authors have made their stance clear in their response that no additional data will be included.   The experimentation is correct, it is how important the data is for figuring out physiological and pathological principles that is modest and still remains modest.  Do we understand why the macaque intestinal mycobiome is different from humans?  No.  Do we understand for humans or any primate what the driving forces are for the mycobiome?  No and this is discussed insuffiently.  This influences the use of this non human primate for study of the human intestinal mycobiome regulation.  The same is true for rodent models and there are many examples of similar physiological events mediated by different bacteria, viruses, or mycotic organisms but different genus and species in rodents versus humans. Tin rodents, the knowledge is being elucidated for the bacterial intestinal microbiome in various longitudinal regions where different physiological events take place.  Some is known about the factors driving the intestinal viral microbiome, but considerably less.  I will credit the authors for conservative writing, that they acknowledge the limited impact of the data as it presently stands.  The decision must come from the editors on whether yet another intestinal  microbiome study should be published and in many cases only observational or correlative and not causative factors of importance of the microbiome are investigated. 

Author Response

We believe that the only valid comment referring to the differences between NHP and human mycobiomes has been covered in the Discussion section of the second revision. The remainder of the reviewers' comments reflect his/her personal opinion about the merits of the manuscript.